# Lipase-Catalyzed Production of Sorbitol Laurate in a “2-in-1” Deep Eutectic System: Factors Affecting the Synthesis and Scalability

**DOI:** 10.3390/molecules26092759

**Published:** 2021-05-07

**Authors:** André Delavault, Oleksandra Opochenska, Laura Laneque, Hannah Soergel, Claudia Muhle-Goll, Katrin Ochsenreither, Christoph Syldatk

**Affiliations:** 1Technical Biology, Institute of Process Engineering in Life Sciences II, Karlsruhe Institute of Technology, 76131 Karlsruhe, Germany; o.opochenska@gmail.com (O.O.); laura.laneque@gmail.com (L.L.); katrin.ochsenreither@kit.edu (K.O.); christoph.syldatk@kit.edu (C.S.); 2Institute for Biological Interfaces 4 and Institute of Organic Chemistry, Karlsruhe Institute of Technology, 76021 Karlsruhe, Germany; hannah.soergel@kit.edu (H.S.); claudia.muhle-goll@kit.edu (C.M.-G.)

**Keywords:** glycolipid, sugar alcohol, ester, biosynthesis, optimization, unconventional media

## Abstract

Surfactants, such as glycolipids, are specialty compounds that can be encountered daily in cleaning agents, pharmaceuticals or even in food. Due to their wide range of applications and, more notably, their presence in hygiene products, the demand is continuously increasing worldwide. The established chemical synthesis of glycolipids presents several disadvantages, such as lack of specificity and selectivity. Moreover, the solubility of polyols, such as sugars or sugar alcohols, in organic solvents is rather low. The enzymatic synthesis of these compounds is, however, possible in nearly water-free media using inexpensive and renewable building blocks. Using lipases, ester formation can be achieved under mild conditions. We propose, herein, a “2-in-1” system that overcomes solubility problems, as a Deep Eutectic System (DES) made of sorbitol and choline chloride replaces either a purely organic or aqueous medium. For the first time, 16 commercially available lipase formulations were compared, and the factors affecting the conversion were investigated to optimize this process, owing to a newly developed High-Performance Liquid Chromatography-Evaporative Light Scattering Detector (HPLC-ELSD) method for quantification. Thus, using 50 g/L of lipase formulation Novozym 435^®^ at 50 °C, the optimized synthesis of sorbitol laurate (SL) allowed to achieve 28% molar conversion of 0.5 M of vinyl laurate to its sugar alcohol monoester when the DES contained 5 wt.% water. After 48h, the *de novo* synthesized glycolipid was separated from the media by liquid–liquid extraction, purified by flash-chromatography and characterized thoroughly by one- and two-dimensional Nuclear Magnetic Resonance (NMR) experiments combined to Mass Spectrometry (MS). In completion, we provide initial proof of scalability for this process. Using a 2.5 L stirred tank reactor (STR) allowed a batch production reaching 25 g/L in a highly viscous two-phase system.

## 1. Introduction

Glycolipids and, more generally, surfactants are getting more attention and are currently under scrutiny from the scientific community because of their various fields of applications [1]. This versatility allows them to fill different functions, such as excipient in drugs, encapsulating agents, lubricants and many more [2]. Such quasi-omnipresence makes their use valuable in very concrete and industrialized applications such as oil recovery enhancement or even as dough conditioning [3,4]. Moreover, the recent sanitary crisis, which resulted from the current global pandemic, is announced to increase further this production, as surfactants are among the main components present in hygiene products [5]. Thus, the whole chain of production starting from the acquisition of raw materials to the formulation and storage should be logically stimulated as well. Given the context, it appears topical to develop processes that enable safe, reliable and straightforward production of such compounds. In addition, the processes and the resulting compounds should respond to present concerns in relation to sustainability, renewability, and biodegradability. Such problems brought to awareness and theorized by the Green Chemistry field revolve around the “benign by design” concept [6]. By definition, it means that chemical products and processes should be designed to eliminate the generation of hazardous substances [7]. Therefore, it reduces the number of steps necessary for the production, from the synthesis to the downstream processing (DSP), and simplifies the overall process chain.

Glycolipids, such as sugar esters (SEs), are, in this regard, relevant candidates that meet the aforementioned requirements; several challenges and problems are raising in the meantime, which are tackled in this work.

The well-established chemical production of SEs presents several disadvantages and limitations such as low specificity, low selectivity and use of corrosive reactants [8]. Consequently, alternative ways were pursued to prevent the use of organic solvents, as they represent most of the waste in industrial processes and syntheses [9]. Moreover, they present undeniable limitations for the solubilization of polyols such as sugars. In this aim, the production of SEs in ionic liquids (ILs) [10] and in Deep Eutectic Systems (DESs) [11] were investigated. The latter, described first by Abbott et al., represents a cheaper, less toxic, and facile option among the low-transition temperature mixtures [12]. Additionally, their low water contents and low water activities lead lipases to reverse their activity and form ester bonds using relatively mild and harmless conditions [13].

Numerous applications of DESs in environmentally friendly chemical processes have been made, notably for the production of pharmaceutically relevant building blocks and scaffolds [14,15,16]. Moreover, DESs can also be used for the extraction of valuable compounds such as phenolics contained in olive oil wastes [17], adding them as a sustainable tool for food waste treatment and re-valorization. In our similar and inherently benign approach, sorbitol is simultaneously part of the solvent as a DES partner, in combination with choline chloride, and part of the lipase-catalyzed reaction as a substrate. Intrinsically, this reaction design solves solubility problems of the polyols while saturating the system with the acyl acceptor. This “2-in-1” principle described first by Siebenhaller et al. [18], using notably C4 to C6 sugars in combination with vinyl fatty esters, presented however exceptionally low yields (~4%) and lacked proof of scalability [19]. The use of vinyl esters for SE synthesis, described by Bornscheuer et al., is thermodynamically favored as the vinyl adduct is converted during the reaction to the side-product ethenol, which itself tautomerizes in the highly volatile acetaldehyde [20] (Figure 1). For the first time, we enhanced further the conversion yield of this advantageous system, using a relatively better acyl acceptor for the reaction [21]. Indeed, sorbitol and, more generally sugar alcohols, showed successful use in prior research as they are also less sensitive to degradation than their actual sugar analogues [22,23,24,25]. Moreover, laurate monoesters are valuable for many recently developed applications such as drug encapsulation [26].

We initiated, herein, the optimization of the “2-in-1” process by selecting Novozym 435^®^ as a suitable biocatalyst among various commercially available lipase formulations and sought, specifically to our system, its optimal performing parameters. We varied parameters (One-Factor-at-a-Time, OFaaT) one at a time, such as biocatalyst concentration, time of reaction, substrate concentration and water content. The optimal temperature for the use of this specific lipase has been extensively studied in literature. Moreover, previous work from Siebenhaller et al. [27] and Hollenbach et al. [28] presented concomitantly that in the range of 50–60 °C, the enzyme meets its optimal activation energy.

In addition, we present robust purification and identification strategies promoted by suitable analytical but also flash-chromatography methods for the quantification and preparative scale (>500 g) production of sorbitol-6-*O*-laurate using a batch stirred tank reactor (STR). The successful investigation was insured by the use of relevant Evaporative Light Scattering Detection (ELSD) combined with HPLC and flash-chromatography. Thus, the production of pure standards enabled the establishment of the calibration curve and the 1D-, 2D-Nuclear Magnetic Resonance (NMR) and Mass Spectrometry (MS) characterization of selectively-acylated sorbitol monolaurate.

## 2. Results

In the following sections, we report the investigation of commercially available lipase formulations to produce SL alongside a newly developed quantification method that allowed the investigation of the impact of several synthesis factors. Following the product formation over time also allowed optimization of reaction time, enzyme and substrate concentration and subsequently, the water content of the DES. Finally, the titer of the optimized process, as well as proof of scalability and technical notes for the use of a batch STR followed by the DSP procedure, are presented.

### 2.1. Sorbitol Monolaurate (SL) Quantification

We, herein, successfully separated and quantified SL using a newly developed analytical HPLC-ESLD method, which allowed the differentiation of the products and substrates (Appendix A). Sorbitol had a retention time of 2.1 min, 3.1 min for SL and ~9.5 min for lauric acid (not integrated). Due to the low baseline noise and suitable peak resolution (Table 1), it was possible to quantify SL in a range between 0.75 g/L and 30 g/L with the use of two linear ranges of calibration as ELSD does not provide a linear response when analyte concentration increased.

### 2.2. Commercially Available Formulations Screening

Figure 2 shows that the highest yields of vinyl laurate conversion into SL were obtained by Lipase B from *Candida antarctica* (CA) formulations such as Novozym 435^®^, Lipozyme 435^®^ and Lipozyme CALB L (liquid formulation). Regardless of the applied form, all of them seem to reach statistically equivalent results of ~20% under non-optimized reaction conditions. However, it is important to note that liquid formulations potentially contain more catalytic material than the immobilized ones, showing the limitations of our comparison.

As we compare these lipase formulations on the basis of applied concentrations (20 g/L), the quantity of actual enzyme present in the reaction most likely differs; moreover, very little data on the matter are made available by the producers. However, the quantity of enzyme is directly linked to the amount of formulation used; thus, when comparing immobilized enzymes, this variable is intrinsically highlighted. As a result, we are, among other factors, comparing qualitatively the immobilization efficiency of the various formulations tested. In this aim and in our system, either using the food-grade formulation (Lipozyme 435^®^) or the technical grade one (Novozym 435^®^) did not lead to significant differences in titer after 48 h of reaction (Table 2). Interestingly the Cross-Linked Enzyme Aggregate (CLEA) from CA seemed to reach statistically similar levels of performance. Therefore, in the following sections, we chose to report the influence of several factors affecting our system. For this purpose, we used Novozym 435^®^ as the archetype of the commercially available lipase formulation for SE synthesis. As a first step, we decided to investigate the evolution of SL titer over time.

### 2.3. Product Formation over Time

To determine the optimal reaction time to produce the maximum amount of SL, the concentration of produced glycolipid was recorded over 96 h. Thus, we reached a saturation of product after 48 h with apparently no significant changes in concentration, even up to 96 h of reaction, as shown in Figure 3.

Under these unoptimized conditions using Novozym 435^®^, a concentration of 95 ± 2.5 mM of the product was obtained after 48 h of reaction, which translates to specific productivity of 100 µmol/h/g. Until now, no data has been reported regarding the use of a “2-in-1” DES system for lipase-catalyzed transesterification between a vinyl laurate and sorbitol in order to synthetize such monoacylated sugar alcohol ester.

### 2.4. Effect of Enzyme, Substrate and Water Contents

Herein, Novozym 435^®^ was used as the biocatalyst for the reaction that was carried out in the “2-in-1” Sorbit DES. Factors impacting the reaction (i.e., enzyme concentration, enzyme dosage and water content in the media) have been investigated to find the optimal parameters for each condition. From Figure 4, it can be discerned that the optimal values were 50 g/L enzyme, 0.5 M vinyl laurate and 5 wt.% water.

Statistical analysis of the different titers reached after 48 h in relation to enzyme concentration revealed significant differences when the amount of enzyme was at least doubled from the 20 g/L starting point (Figure 4A). Drastic differences can be observed in Figure 4B, as 0.5 M of vinyl laurate induces a two-fold increase in titer compared to 0.25 M and 0.75 M. This safely suggests that upon a range of concentration >0.5 M, there was potentially an inhibition of the biocatalyst due to substrate saturation. Above this concentration and in spite of the statistical analysis, comparisons are rendered ambiguous considering the lack of trend and should be interpreted with a certain distance. The results displayed in Figure 4C are rather more ambiguous as well, concerning the water content. Indeed, above 5 wt.% water, it seems that an optimal range of water content was reached. Overall, the results tend to indicate that enzyme and substrate concentrations were predominant factors compared to the water content, to some extent, as the Sorbit DES with a 2.5 wt.% water content ultimately limits the conversion most probably due to its measured 2-fold higher viscosity (Table A1). It also seems that the Sorbit DES with 5 wt.% water provides the optimal A_w_ for the enzymatic activity (~0.08). Thus, a compromise has been found on the water content of the Sorbit DES, giving a good balance between viscosity and water activity for an optimal substrate conversion. In the following section, we display the structural elucidation of the product and the impact of the optimized combined factors on our process.

### 2.5. Structural Elucidation Using Spectroscopic and Spectrometric Methods

Here, we report one of the most extensive structural characterizations of SL. ^1^H- and ^13^C-Nuclear Magnetic Resonance (NMR) Spectroscopy and Mass Spectrometry (MS) were performed on the purified compound. Additional ^1^H-^1^H COSY and ^1^H-^1^H TOCSY, ^1^H-^13^C HSQC and ^1^H-^13^C HMBC (Appendix A) combined with MS results, confirmed that only one acyl group (laurate adduct) was bound to a primary hydroxyl group of sorbitol. From the 1D- and 2D-NMR experiments, the chemical shifts and their assignments are fully detailed in Table 3.

To determine which primary carbon of the *meso-* and the pseudo-asymmetric polyol is bound to laurate, we measured a ^1^H-^13^C-HSQC with high resolution in the direct dimension and no decoupling. Extraction of 1D slices along the respective carbon frequency allowed estimation of vicinal ^1H-1H^J couplings. Figure A1 shows the signals that belong to C^4^H and C^3^H of the sorbitol. C^4^H is a triplet with coupling constants of 4.8 Hz, indicating that both vicinal protons to this group are in *cis* position. C^3^H, on the other hand, displayed a doublet of doublets with two different coupling constants of 5.3 Hz and 7.4 Hz; thus, here, the vicinal protons are different, best explained by one in *cis* and the other in *trans* position. This proves that the laurate adduct is connected via the ester function to the C^1^H end of the polyol.

MS was performed with ElectroSpray Ionization (ESI) (Appendix A). The spectrometric analysis of the synthesized SL is shown below in Table 4. In the latter, 5 clear adducts of SL have been observed and correlated to possible adducts.

### 2.6. Optimized Tube Scale Production and Scalabilty

Combining the optimized factors allowed a product concentration of up to 50 g/L of SL at the tube scale to be reached. Interestingly, to demonstrate the scalability of the production in a batch STR, we ended up with a ~2-fold decrease of both titer and yield, as shown in Table 5, suggesting that the reaction is sensitive to the effect of the reactor. Presumably, the chosen parameters such as stirring speed and stirring did not match the performance of orbital shaking and resulted in lower performance. Albeit we demonstrated in the present work the scalability of our process. More investigation specific on this reactor system that would focus on these parameters is needed.

In Figure 5, we observe directly this blatant two-fold decrease in titer. However, it is interesting to notice an equivalent titer in the early stage of the reaction being reached in the STR and the orbitally shaken tubes with comparable initial velocities. After 24 h, the difference in titer increases drastically while the specific productivity dropped in both reactors with a 1:2.5 ratio between the two systems. Finally, at 48 h, reaction velocities decreased again by roughly 2-fold, conserving a similar ratio as previous. Thus, it is suspected that, comparatively, the enzyme was inhibited due to mass transfer limitations in the STR.

We propose thereafter a discussion making a parallel between our results and pre-existing ones in literature. Allowing us to give a broader aspect of the role of immobilized enzymes, DESs and reactor technologies in sustainable processes such as, but not only limited to, sugar ester synthesis.

## 3. Discussion

### 3.1. Immobilized Lipases and Media Tailoring for Sustainable Biocatalyzed Esterifications

To enhance the performance of the biocatalyst, the immobilization of enzymes and, more specifically, lipases are of high industrial interest, and it has become a requirement [29]. Indeed, it was shown in numerous reports that through, e.g., adsorption or covalent binding with a solid carrier, some acyltransferases, such as the lipase B from *Candida antarctica* (CaLB), could display improved activity, stability and reusability [30,31,32]. This phenomenon can be understood *inter alia* as a gain of rigidity for the enzyme’s tertiary structure, which limits protein unfolding and thus denaturation of the biocatalyst [33]. This is in practice true in the case of the Novozym 435^®^ formulation that was recently qualified as the “perfect immobilized biocatalyst” in a review from Ortiz et al. [34]. In parallel, the native CaLB has been demonstrated to be among the most stable commercially available lipases [35,36]. By binding it to a macroporous acrylic polymer resin (Lewatit VP OC 16,000), resulting in the now highly reported Novozym 435^®^, it increased drastically the performances of the biocatalyst [37].

In the current work, it might appear shocking that we use 50 g/L of the enzyme to reach the same value in titer of product. At least on the tube scale, it is likely that, in the case of Novozym 435^®^, the costs associated with the immobilization might be fully compensated [38,39,40,41,42,43,44,45,46,47,48]. Indeed, the formulation can be recycled, using filtration, for sometimes up to 10 cycles, as a report from Liu et al. demonstrates [49]. It is also important to remember that in such a bead-adsorbed enzyme formulation, the non-catalytic ballast of the carrier represents a tremendous portion of the biocatalyst’s mass (90 to >99 wt.%) [37,50]. Thus, it is remarkable that in our work, the immobilized versions of CaLB, such as Lipozyme 435^®^ or Novozym 435^®^, rivalized with the buffered formulations as they should normally contain a higher enzyme concentration.

Interestingly, CLEA CA was, in our case, comparable to the other CaLB formulations. The Cross-Linked Enzyme Aggregates (CLEAs) first introduced by Sheldon et al. [51], but also the Cross-Linked Enzyme Crystals (CLECs) [52], represent very interesting and more elegant alternatives to the petrol-based polymer-carriers. CLEAs are a result of firsthand the precipitation and physical aggregation of the enzymes, then secondhand the cross-linking of these aggregates with a cross-linking agent, which can be typically glutaraldehyde. This gives several advantages to CLEAs over their non-covalently immobilized analogues, such as the quasi-nonexistent leaching of enzyme even under reportedly harsh conditions [53]. Due to the covalent inter-molecular binding, CLEAs and CLECs afford complete removal of the carriers, thus resulting in carrier-free immobilized enzymes. However, they also present limitations and challenges for their industrial-scale production, notably for the control of the enzyme’s aggregation that can result in less active enzyme dimers [54]; thus, they are, to this day, rather rarely produced in bulk. On a side note, it is important to mention that standard acrylic-bead-immobilized enzymes were easier to handle throughout the tube-scale synthesis and downstream processing, than their CLEA analogues.

Herein, we reached an optimal titer after 48 h of reaction, which is like other reports dealing with enzymatic production of sugar esters in DESs [11] but much shorter than microbial fermentation to produce glycolipids. Indeed, as an example, to produce Mannosylerythritol Lipids (MELs), the average harvesting time is between 7 to 10 days for a titer ranging from 15 g/L to 165 g/L, to obtain, in the case of the highest yields, complex mixtures of MELs [55,56,57,58]. Additionally, and as we demonstrated, fewer factors must be investigated when selectively producing tailor-made glycolipids using either free or immobilized enzymes. On the other hand, microorganisms, such as yeasts or fungi, exhibit much more complex behaviors that require acute control of the reaction conditions. In this specific case, and despite MELs being well established on the market, mixtures of products are often obtained, thus adding a degree of complexity to the DSP. Our process using a “2-in-1” DES as media is rather more straightforward comparatively, but more time and investigation are needed to overcome challenges to make it relevant for industrial production. In regard to DSP, it has been shown in several studies that DESs can be recycled as well [59,60,61]. In our case and for an efficient liquid–liquid extraction, the “2-in-1” DES was first disrupted via aqueous solvation. Thus, it could be foreseeable that choline chloride-rich wastes generated by such DES-mediated process could be re-valorized in feed additives [62] or as agrochemical active ingredients [63]. Unlike organic solvents, DESs do not have to undergo incineration; thus, their release in nature can be considered [64].

Analogously, a eutectic mixture using organic solvents (NaOH/DMSO/2M2B) was investigated by Kim et al. for the synthesis of SL using lauric acid, reaching exceptionally high yields (97%) using ~500 g/L of the enzyme [23]. Despite this remarkable achievement, we propose, in contrast, a method using a deep eutectic system made of ubiquitous, renewable and inexpensive compounds, such as choline chloride and sorbitol, using 10 times less biocatalyst. Concomitantly, they also reported an adverse effect of highly viscous mixtures on the reaction, which correlates to the decreased titer we obtained when the reaction’s water content was set to 2.5 wt.%. Overall, this might also explain our comparatively inferior conversion yields, as it seems that deep eutectic systems present challenges in terms of mass transfer limitations, meaning that our substrate hardly moves to the enzyme’s active site. Zhao et al. reported a bisolvent system containing either ILs or DESs in combination with 2M2B for glucose-based fatty acid esters production. Similar factors were investigated and gave results equivalent to ours, such as an optimal enzyme content of 20 g/L of Novozym 435^®^ at 60 °C to reach 46 % conversion yield from the 0.3 M of vinyl ester used [65].

The ever-growing knowledge on enzyme immobilization and media-tailoring technologies have shown to be crucial tools to reach sustainability in biocatalysis. They are also of active interest for various industrial domains to develop greener and sustainable processes [18,66,67,68,69]. Regarding this aim and as performance is of keen interest for industrial application, reactor technology and scalability represent two important pillars as well.

### 3.2. Reactor Technology: Toward Scaling-Up Lipase-Catalyzed Reactions

The development of suitable reactors and technologies that support lipase-catalyzed reactions is a topical subject for both academia and industry. Some parameters inherent to reactors have been shown to be of high importance for the performance of the process. For example, the speed of stirring can drastically influence the initial velocity of the reaction and the conversion yield. This was clearly demonstrated in a research article from Korupp et al. dealing with the enzymatic production of glycerol adipate using Novozym 435^®^ [70]. In the latter, they observed the highest conversion rates at 100 rpm. The type of stirrer did not significantly influence the reaction; however, it was observed that only a helicon ribbon stirrer gave uniform convection of both substrate and catalyst. A similar observation was made in our case using a 3-bladed spiral propeller instead of a 4-bladed flat turbine (unpublished data), suggesting that homogeneity of highly viscous mixtures can be reached optimally in an STR with stirrers that induce vertical convection due to a rather axial flow. Indeed, studies from Wiemann et al. and Ansorge-Schumacher et al. are suggesting that viscous mixtures requiring high energy input might have a deleterious effect on the physical and mechanical properties of immobilized formulations [71,72]. To corroborate these affirmations, a study from Keng et al. used a Rushton turbine impeller, providing this time a radial flow that was seemingly more adapted to the relatively lower viscosities of their *n*-hexane-based mixture [73]. In their report dealing with the Lipozyme RM IM-catalyzed palm esters synthesis, it was clearly concluded as well that the shear effect of high impeller speed on the enzyme particles caused an adverse effect on reaction performances. Thus, potentially explaining our significant loss of titer (~2-fold decrease) and lower reaction velocity when scaling up from the tube to the STR (Table 5 and Figure 5). Altogether, it seems that a compromise needs to be found on the stirrer type and its speed to conserve the integrity of the biocatalyst. In this regard, more stable lipase formulations are commercialized to respond to this problem. A good example is the recently co-developed and industrially produced CalB immo Plus^®^ from c-Lecta and Purolite companies (www.c-lecta.com, Leipzig, Germany/www.purolite.com, King of Prussia, PA, USA). The highly hydrophobic carrier ECR1030M (DVB/acrylate copolymer) exhibits enhanced mechanical stability and offers a controlled size of spherical beads.

As a matter of fact, we mostly demonstrate what conditions were the most intrinsically influential on the reaction, but further studies would be needed to optimize the process at an even bigger scale. We essentially brought in the current work proof of the scalability of our process and, more generally, the scalability of the DES-mediated biocatalysis, which is scarcely represented in the literature [74]. Despite a lack of in-depth understanding of how mechanistically DESs can activate and stabilize lipases, the room for improvement of this topic toward industrial application is rather wide. However, we were able to demonstrate that the preparative scale is reachable using a facile and straightforward method that requires the minimum necessary equipment, as shown in the flowsheet of Figure 6. We removed, therein, the need for continuous pH and gas composition assessment that are standardly used in microbial fermentation, among other measurements that require probes combined with their respective computer software.

Although our discussion mainly revolved around the use of rather conventional reactors, such as STRs for batch heterogenous biocatalysis, innovative and commercially available alternatives have risen on the market. The Rotating Bed Reactors (RBRs), notably commercialized by the SpinChem company (www.spinchem.com, Tvistevägen, Sweden), have proven to be promising alternatives to STRs and have shown good results when used for immobilized enzymatic reactions [75]. Furthermore, their cooperation with Purolite (www.purolite.com, King of Prussia, PA, USA) gave light to immobilized lipase cartridges, theoretically forming a rotating packed bed reactor, which removes the need for filtration during the DSP and simplifies the recovery of the biocatalyst. Remarkably, packed bed reactors have also been combined with DESs for lipase-catalyzed esterifications. In a relatively recent study from Guajardo et al., they managed the transition from a batch to a fed-batch and continuous process, enhancing simultaneously conversion yield and productivity, thus seemingly resolving the encountered issue of mass transfer limitations and biocatalyst inhibition [76].

Recently as well, our research group published proof of a concept for the simultaneous extraction of lipids from yeast and the subsequent ester production in a “2-in-1” Sorbit DES, using microwave heating as an alternative to thermal heating [25]. This simplified and fast method is only an example of the vast possibilities that DESs, immobilized enzymes and innovative reactor systems can offer not only to the field of biocatalysis but also to biotechnology in a broader scope.

## 4. Materials and Methods

### 4.1. Materials

Vinyl laurate was purchased from Tokyo Chemical Industry Co., Ltd. (TCI Europe, Zwijndrecht, Belgium). Lipase formulations: Novozym 435^®^, Lipozyme 435^®^ and Novozym NS 81356 were given by Novozymes (Bagsværd, Denmark). CalB Immo Plus^®^ was given by c-Lecta (Leipzig, Germany). CalA Immo 150, Lipase TL CLEA, Lipase CA CLEA and all other chemicals were purchased from either Carl Roth GbmH & Co. KG (Karlsruhe, Germany) or Sigma Aldrich Chemie GmbH (Taufkirchen, Germany) if not stated otherwise. The other 9 lipase formulations were acquired in the Novozymes Lipase Screening Kit purchased from Strem Chemicals (Newburyport, MA, USA).

### 4.2. DES Preparation and Standard SL Synthesis for Enzyme Formulation Screening

The sorbitol and choline chloride-based DES, dubbed “Sorbit DES”, was prepared and validated according to the procedure described by Dai et al. and Hayyan et al. [77,78]. The water content was varied (1.25, 2.5, 5, 7.5 and 10 wt.%) then controlled according to the method described in Section 4.7.

In a 5 mL Eppendorf tube, were introduced subsequently 1.5 mL of warm Sorbit DES, vinyl laurate (195 µL, 170 mg, 0.75 mmol, 0.5 M) and 30 mg of enzyme formulation (20 g/L) (Table 1). The tubes containing the reaction mixture were agitated in a rotator and vortex mixer (program U2) from neoLab (Heidelberg, Germany) at 90 rpm and 50 °C. To get a triplicate for each measure, three tubes were collected for each time point at: 0.5, 4, 8, 24, 28, 32, 48, 72 and 96 h. The latter were then processed for further analysis as described in Section 4.8, the conversion yields were calculated as the percentage of molar ratio of sorbitol monoester produced to the total amount of vinyl laurate added to the reaction system.

### 4.3. Enzyme Formulations Screening

To compare the lipase formulations (Table 1), an identical concentration of the latter was used each time (20 g/L), and the same substrate concentration (0.5 M vinyl laurate) was provided to the media at 50 °C. For this comparison, the reaction was stopped after 48 h.

### 4.4. Influence of Enzyme Concentration

In order to examine the optimal concentration of lipase formulation, different concentrations of Novozym 435^®^ (10, 20, 30, 40, 50 and 60 g/L) were tested without varying any other reaction parameter.

### 4.5. Optimization of Vinyl Laurate Amount

To address the optimal vinyl fatty ester concentration for the reaction, different vinyl laurate concentrations (0.25, 0.5, 0.75, 1 and 1.25 M) were tested. All other reaction conditions were kept constant.

### 4.6. Water Content, Viscosity and Water Activity Analysis

The water content was assessed with Karl-Fischer titration using a TritoLine 7500 KF trace from SI Analytics (Mainz, Germany) at 20 °C in combination with Aquastar CombiCoulomat fritless (Merck Millipore, Darmstadt, Germany) as analyte. Water standards of 0.1 and 1% in xylene from Merck Millipore (Darmstadt, Germany) were used to test the titrator’s accuracy before the measurements.

The viscosity was measured at 50 °C with a viscosimeter MCR 501 using a CC10 concentric cylinder (Anton-Paar, Graz, Austria) with about 1 mL of liquid for each water content for the Sorbit DES.

The water activity was measured at 50 °C with a LabMaster-aw neo A_w_meter (Novasina, Lachen, Switzerland) using 3 mL of liquid for each water content for the Sorbit DES.

### 4.7. Scale-Up Procedure and Downstream Processing for Standard and Bulk SL Production

In a 2.5 L Minifors bioreactor (Infors HT, Bottmingen, Switzerland), 500 mL of warm Sorbit DES were introduced, prepared as described in Section 4.2. The medium temperature was first equilibrated to 50 °C then subsequently were introduced 25 g of Novozym 435^®^ (50 g/L) and vinyl laurate (65 mL, 56.62 g, 0.25 mol, 0.5 M). The reaction mixture was stirred at 300 rpm with a single three-bladed spiral propeller (D = 54 × 12 mm) (Infors HT, Bottmingen, Switzerland). After 48 h, the reaction was stopped, the media diluted with 500 mL of double-distilled water and filtrated through a Büchner funnel. In total, 200 mL of brine were incorporated into the aqueous phase that was then extracted 6 times with a 1:1 volume ratio of ethyl acetate. The organic phases were gathered and chemically dried over MgSO_4_ before being dry evaporated with a rotative evaporator.

For the purification of the SL standard that was used for the calibration curve, 2 g of the crude paste was re-dissolved in chloroform to be adsorbed over 4 g of Celite 545 for flash chromatography purification using the solid loading method. To purify this crude, a Reveleris PREP purification system equipped with a 12 g Chromabond^®^ Flash RS 15 Sphere SiOH column (Macherey-Nagel, Düren, Germany) was used. Elution solvents were chloroform and methanol with a gradient such as 2nd solvent percentage started at 0% for 1.5 min, 7% for 9.5 min, 15% for 3 min and finally 100% for 3 min. The second fraction, containing SL, was collected at 7.5 min. The latter was dry evaporated on a rotary evaporator for further use and analysis; ~0.75 g of dried, white powder (Appendix A).

For bulk production of SL, the entire crude (~25 g) was re-dissolved in the necessary minimal volume of ethyl acetate (~400 mL) and washed with 1 × 400 mL of brine, and then 3 × 400 mL of double-distilled water to remove unreacted sorbitol. The organic phase was dried over MgSO_4_ and dry evaporated on a rotary evaporator; ~10 g of dried, white powder (Appendix A).

### 4.8. Sample Preparation and HPLC-ELSD Quantification Method

Tubes produced by the methods described in Section 4.2, Section 4.3, Section 4.4 and Section 4.5 were prepared and analyzed as follows. For extraction of the glycolipid and its quantification with HPLC-ELSD, the following procedure was applied. First, 1 mL of double-distilled water was added to the tube containing the reaction mixture and vortexed for 45 s. Then, 2.5 mL of ethyl acetate was added to the warm solubilized DES; subsequently, the extraction took place at 50 °C for 20 min with the use of the orbital shaker set on program U2 and 90 rpm (neoLab, Heidelberg, Germany). Then, 1 mL of the upper phase was aliquoted and dried on a centrifugal evaporator in order to be resolubilized in 1 mL of chloroform/methanol (75:25 *v*/*v*), and 100 µL was placed into an HPLC vial for further quantification.

The method described by Hollenbach et al. [28,79] was used with slight modifications as follows. Kinetex EVO C18 (2.6 mm, 250 mm × 4.6 mm) column from Phenomenex (Aschaffenburg, Germany) with an accompanying guard column (4 mm × 3.0 mm ID) of the same phase, using an Agilent 1260 series liquid chromatograph (Waldbronn, Germany) equipped with a quaternary pump, an autosampler and a column oven. An evaporative light scattering detector (ELSD) from BÜCHI Labortechnik (Essen, Germany) was used for detection. The mobile phase, solvent A, was water and solvent B was acetonitrile. The flow rate was 1 mL/min, and a gradient was used for separation of product and substrates: starting from 40% A–60% B, then 0–10 min a linear gradient up to 35% A–65% B, followed by another linear gradient from 10 to 15 min up to 25% A–75% B. This gradient was held for 5 min, followed by a reconditioning step of the column with 40% A–60% B for 5 min. The injection volume was set to 10 µL. The column was operated at 50 °C. The detector was operated at 38 °C with a gas flow (air) of 1.5 mL/min. The gain was set to 1. The retention times were 2.1 min for sorbitol, 3.5 min for SL and ~9.5 min for lauric acid (Appendix A).
(1)$$\mathrm{Yield}\;\left[\%\right]=\frac{\mathrm{nSL}\;\left[\mathrm{mol}\right]}{\mathrm{nVinyl}\;\mathrm{laurate}\;\left[\mathrm{mol}\right]}\times 100\%.$$
(2)$$\mathrm{Specific}\;\mathrm{reaction}\;\mathrm{velocity}\;\left[\text{µ}\mathrm{mol}/h/g\right]=\frac{\mathrm{nSL}\;\left[\text{µ}\mathrm{mol}\right]}{\mathrm{mBiocatalyst}\;\left[g\right]\times \;\mathrm{time}\;\left[h\right]}$$
n: number of moles.m: mass.

### 4.9. Spectroscopic and Spectrometric Methods for Structural Elucidation of SL

For nuclear magnetic resonance (NMR) spectroscopy, 10 mg of purified SL was dissolved in 0.6 mL CD_2_Cl_2_/*d*_6_-acetone (4:1, by vol.). Then, 1D ^1^H-NMR spectroscopy and 2D ^1^H-^1^H correlation spectroscopy (COSY), ^1^H-^1^H total correlation spectroscopy (TOCSY), ^1^H-^13^C heteronuclear single-quantum correlation (HSQC) spectroscopy and ^1^H-^13^C-heteronuclear multiple-bond correlation (HMBC) spectroscopy were performed on a Bruker AVANCE III 600 MHz spectrometer (Bruker BioSpin, Rheinstetten, Germany) equipped with a TCI cryoprobe at a temperature of 27 °C. Spectra were processed and analyzed using Topspin 4.0.1 (Bruker BioSpin) and MestReNova 14.2.0 (Mestrelab Research S.L., Santiago de Compostela, Spain). Chemical shifts were referenced to the ^1^H and ^13^C-resonance of tetramethylsilane (TMS).

The mass spectrometry (MS) for mass identifications was performed with electrospray ionization (ESI) on a quadrupole Q Exactive Plus (ThermoFisher Scientific GmbH, Kandel, Germany) and recorded in positive mode, and raw spectrometric data were treated using MestReNova Suite 2020 (version 14.2.0) (Mestrelab Research S.L., Santiago de Compostela, Spain).

### 4.10. Data Treatment and Statistical Analysis

OriginPro software 9.7 (version 2020) (OriginLab Corporation, Northampton, MA, USA) was used for raw data treatment and statistical analysis. Results are presented as mean ± standard deviation (*n* = 3). Statistical analysis was performed by one-way ANOVA and Tukey test, and the results were considered significant if the *p*-value was <0.05 if not stated otherwise.

## 5. Conclusions

In this work, we presented the screening of 16 commercially available lipase formulations for the DES-mediated and lipase-catalyzed production of glycolipid sorbitol-6-*O*-laurate. We determined the influence of several factors, such as time of reaction, enzyme dosage, substrate concentration and water content, when using Novozym 435^®^ as biocatalyst owing to our newly HPLC-ELSD quantification method. We, therein, elucidated and identified, with complete spectroscopy and spectrometry analysis, the structure of the targeted compound. To finalize our report, we brought evidence of the possible scalability of the process and its importance for sustainable biocatalysis by highlighting analogous knowledge and facts from the literature. Overall, we show that despite many challenges and limitations, DESs demonstrate factually potential for mediating bioprocesses. Thus, it is foreseeable that more and more concrete applications will emerge concerning this specific topic.

## Figures and Tables

**Figure 1 molecules-26-02759-f001:**
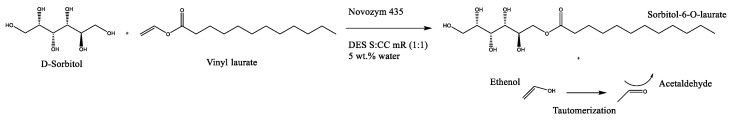
Lipase-catalyzed transesterification reaction between D-sorbitol and vinyl laurate. Evaporation of highly volatile acetaldehyde makes the conversion irreversible. S: Sorbitol; CC: Choline Chloride; mR: molar Ratio.

**Figure 2 molecules-26-02759-f002:**
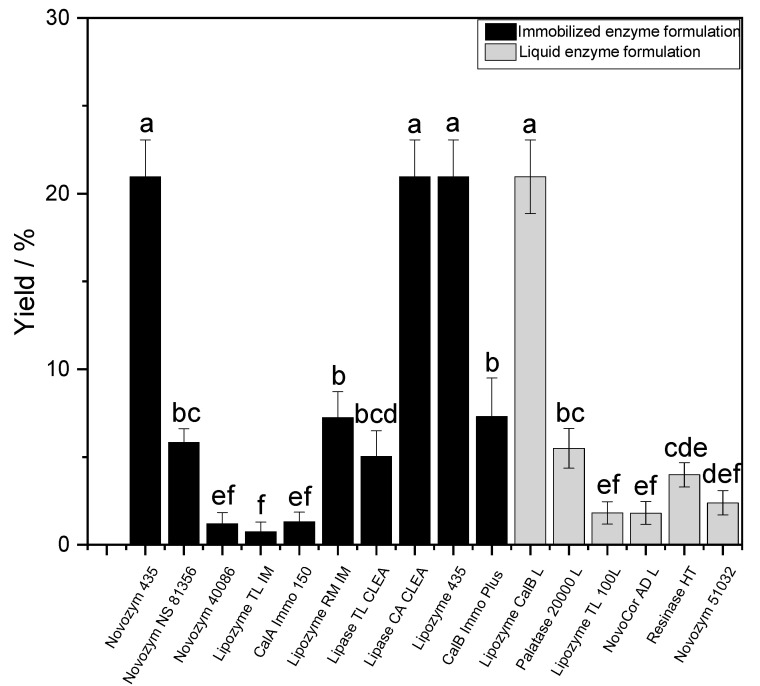
Comparison of SL conversion yields calculated from 0.5 M vinyl laurate. A triplicate was done for each screened commercially available formulation after 48 h at 50 °C. a–f show statistically significant differences (*p* < 0.05).

**Figure 3 molecules-26-02759-f003:**
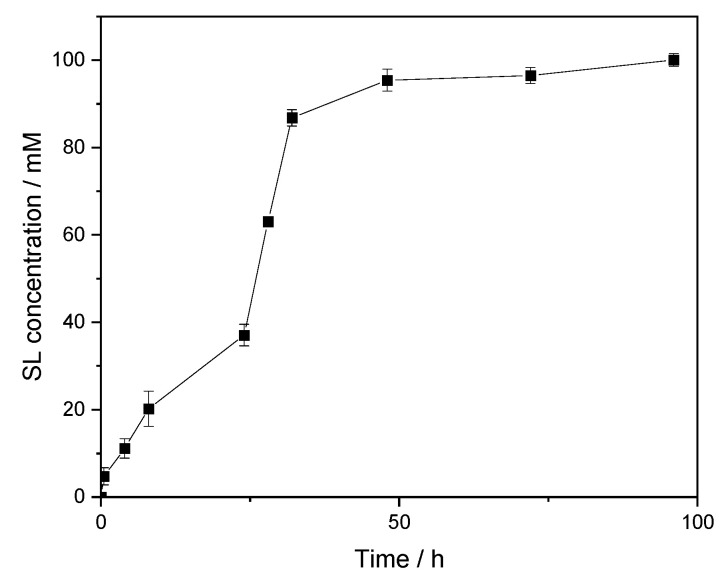
Time course of the reaction under unoptimized conditions: 0.5 M of vinyl laurate, 20 mg of Novozym 435^®^ and 1.5 mL of Sorbit DES as solvent (sorbitol/choline chloride, 1:1, mR, 5 wt.% water) at 50 °C.

**Figure 4 molecules-26-02759-f004:**
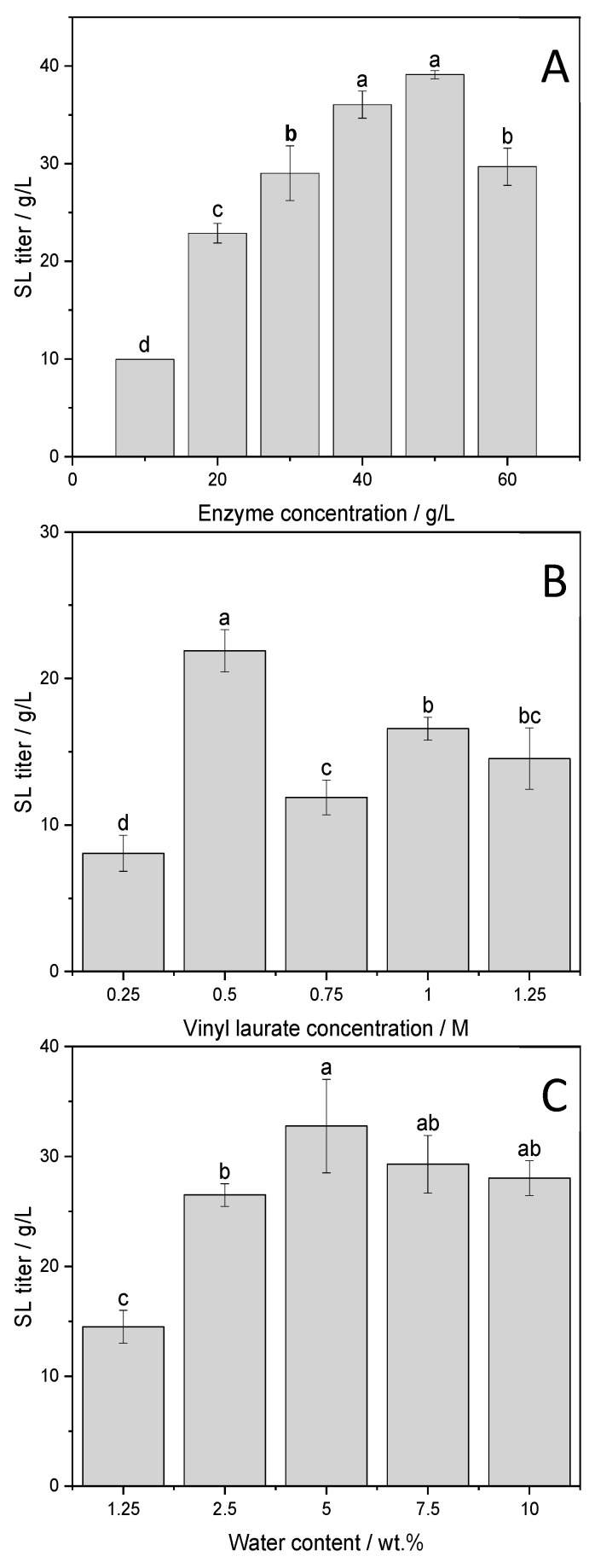
Novozym 435^®^-catalyzed transesterification of sorbitol and vinyl laurate in “2-in-1” Sorbit deep eutectic system: Effect of enzyme dosage (**A**); vinyl laurate concentration (**B**); water content in the media (**C**) on the titer after 48 h. a–d show statistically significant differences, at a 0.05 significance level, of the mean values obtained from three independent experiments ran under each condition.

**Figure 5 molecules-26-02759-f005:**
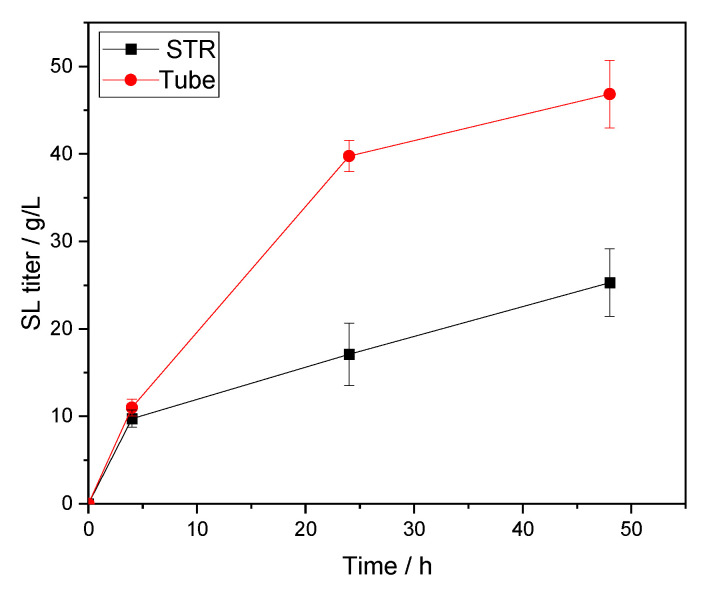
Time course of SL production (g/L): STR with 3-bladed spiral propeller versus tube in orbital shaking.

**Figure 6 molecules-26-02759-f006:**
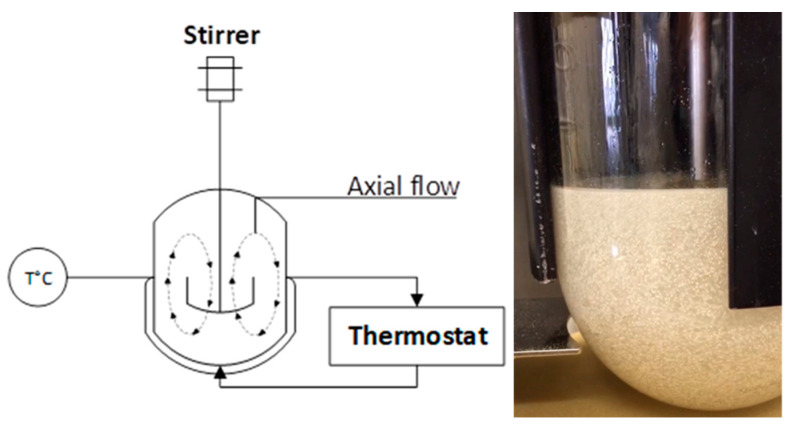
Flowsheet and picture illustrating the visibly homogenized lipase-catalyzed production of sugar alcohol monoesters using a stirred-tank reactor.

**Table 1 molecules-26-02759-t001:** Chromatographic and analytical characteristics of SL analysis using HPLC-ESLD.

Retention time (SL) *	3.55–3.59 min
Peak width **	0.060–0.091 min
Resolution_sorbitol—SL_ (*n* = 3)	17.5
Resolution_SL—lauric acid_ (*n* = 3)	7.6
Baseline noise (*n* =3)	0.22 ± 0.07 mV
Limit of detection (signal/noise = 3)	<0.04 g/L
Limit of quantification (signal/noise = 10)	0.04 g/L
**1st Range of Linear Calibration**	
Correlation coefficient (R^2^, *n* = 3)	0.9967
Equation of linear calibration	y = 0.001x + 0.988
Linear range of calibration	0.75–15 g/L
**2nd Range of Linear Calibration**	
Correlation coefficient (R^2^, *n* = 3)	0.9993
Equation of linear calibration	y = 0.002x − 21.639
Linear range of calibration	20–30 g/L

* Inter-day variance of retention time measured at 3 different days. ** Concentration 0.75–30 g/L.

**Table 2 molecules-26-02759-t002:** Titer (g/L) of SL obtained after 48 h with commercially available enzyme formulations.

Formulation Name	Reported Activity *	Formulation Type	Reported Optimal Temperature Range (°C) *	Titer (g/L) **
CalA Immo 150	500 U/g	Immobilized	N.C.	2.3 ± 0.9
CalB Immo Plus	>9000 PLU/g	Immobilized	60–80	13.3 ± 3.0
Lipase CA CLEA	>1.5 U/mg	Cross-linked	>40	38.2 ± 3.8
Lipase TL CLEA	≥25 U/mg	Cross-linked	40–60	9.2 ± 2.7
Lipozym 435	9000 PLU/g	Immobilized	N.C.	38.2 ± 3.8
Lipozyme CALB L	5000 LU/g	Liquid	30–60	38.2 ± 3.8
Lipozyme RM IM	275 IUN/g	Immobilized	30–50	13.2 ± 2.7
Lipozyme TL 100L	100 KLU/g	Liquid	20–50	3.3 ± 1.2
Lipozyme TL IM	250 IUN/g	Immobilized	50–75	1.3 ± 1.0
NovoCor AD L	6000 LU/g	Liquid	30–60	3.3 ± 1.2
Novozym 40086	275 IUN/g	Immobilized	30–50	2.2 ± 1.1
Novozym 435	10,000 PLU/g	Immobilized	30–60	38.2 ± 3.8
Novozym 51032	15 KLU/g	Liquid	35–70	4.4 ± 1.3
Novozym NS 81356	N.C.	Immobilized	N.C.	10.6 ± 1.4
Palatase 20000 L	20,000 LU/g	Liquid	30–50	10.0 ± 2.1
Resinase HT	50 KLU/g	Liquid	≤90	7.3 ± 1.3

N.C.: Not Communicated. * Reported data are provided by the producers and are available online. ** Experiments were performed as triplicates under identical conditions: 0.5 M of vinyl laurate and 20 mg of formulation in 1.5 mL of Sorbit DES (sorbitol/choline chloride, 1:1, mR, 5 wt.% water) after 48h at 50 °C. Data is presented as mean values *±* standard deviations (*n* = 3, *p*-value < 0.05).

**Table 3 molecules-26-02759-t003:** ^1^H- and ^13^C-NMR chemical shifts of sorbitol-6-*O*-laurate (SL) with their molecular assignments.

Molecular Group	^13^C Shift (ppm)	^1^H Shift (ppm)	Multiplicity	Coupling (Hz)
Sorbitol				
-C^1^H_2_-O- *	66.66	4.35, 4.18	dd, dd	3.0, 11.5, 6.3
-C^2^H-	70.69	3.91	m	-
-C^3^H-	73.24	3.68	dd	5.28, 7.37
-C^4^H-	70.27	3.93	t	4.84
-C^5^H-	74.32	3.83	m	-
OH-C^6^H_2_-	64.25	3.71	m	-
Laurate				
O=C^1^-OH *	174.46	-	-	-
-C^2^H_2_- *	34.64	2.33	t	7.6
-C^3^H_2_-	25.46	1.61	m	-
-C^4^H_2_-	29.72	1.32	m	-
-C^5^H_2_-	30.12	1.28	m	-
-C^6^H_2_-	30.12	1.28	m	-
-C^7^H_2_-	30.12	1.28	m	-
-C^8^H_2_-	30.12	1.28	m	-
-C^9^H_2_-	30.12	1.28	m	-
-C^10^H_2_-	30.12	1.28	m	-
-C^11^H_2_-	23.22	1.29	m	-
-C^12^H_3_	14.35	0.88	t	7.00

* Acylation site; d = doublet; t = triplet; m = multiplet.

**Table 4 molecules-26-02759-t004:** Adducts determined from the observed *m/z* obtained via ElectroSpray Ionization-Quadrupole (ESI-Q) experiment for the purified Sorbitol-6-*O*-Laurate (SL).

Observed *m*/*z* Value	Corresponding Adduct	Relative Abundance (%) *
329.232	[M_SL_ + H]^+^-2H_2_O	7.27
347.243	[M_SL_ + H]^+^-H_2_O	100
365.253	[M_SL_ + H]^+^	34.05
382.280	[M_SL_ + NH_4_]^+^	0.92
387.235	[M_SL_ + Na]^+^	18.57

* Calculated from the abundance of the ionic fragments on the y-axis, which was produced by in source fragmentation.

**Table 5 molecules-26-02759-t005:** Impact of optimized factors on SL titer after 48 h in Sorbit DES with 5 wt.% water, using Novozym 435^®^ at tube and stirred-tank scales. Reaction conditions: 0.5 M vinyl laurate, 50 °C and 50 g/L enzyme formulation.

Scale/Stirring	Titer (g/L)	Yield *** (%)	Specific Reaction Velocity *** (µmol/h/g)		
			4 h	24 h	48 h
Tube/Orbital Shaking	50 ± 3 *	28 ± 2 *	151 ± 13 *	91 ± 4 *	54 ± 5 *
Stirred-Tank/3-bladed spiral propeller	25 ± 8 **	14 ± 4 **	134 ± 10 **	40 ± 8 **	29 ± 4 **

* *n* = 3, *p*-value < 0.05. ** *n* = 2, *p*-value < 0.1. *** Calculated from Equations (1) and (2), Section 4.9.

## Data Availability

The main data presented in this study are available in the Appendix A.

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
