# Peer review of "Lipase-Catalyzed Production of Sorbitol Laurate in a “2-in-1” Deep Eutectic System: Factors Affecting the Synthesis and Scalability"

_molecules, 2021, doi:10.3390/molecules26092759_

Round 1

Reviewer 1 Report

In this work, the authors presented the screening of differents commercially available lipase formulations for the DES mediated and lipase-catalyzed production of glycolipid sorbitol-6-O-laurate. Very interesting is the use of the reactor they support reactions catalyzed by lipase. The manuscript is well written and the results well presented. I would suggest increasing the literature with articles concerning the use of DESs in chemistry, both as reaction media (Biorenewable Deep Eutectic Solvent for Selective and Scalable Conversion of Furfural into Cyclopentenone Derivatives, Molecules, 2018; Green Synthesis of Privileged Benzimidazole Scaffolds Using Active Deep Eutectic Solvent, Molecules, 2019; Deep Eutectic Solvents as Effective Reaction Media for the Synthesis of 2-Hydroxyphenylbenzimidazole-based Scaffolds en Route to Donepezil-Like Compounds. Molecules. 2020;) and as environmentally sustainable solvents in extraction processes (Natural deep eutectic solvent as extraction media for the main phenolic compounds from olive oil processing wastes, Antioxidants, 2020). Also I would suggest, if possible, to test the recycling of used DES and test its reuse by entering sub-paragraph 3.3. After the suggested revisions, the manuscript can be published in Molecules

Author Response

We hereby thank the reviewer for his comments and suggestions.

We deem these suggestions of literature highly relevant for the betterment of the manuscript, thus all of them have been included in the introduction with respective and additional short paragraphs. To support the use of:

  • DESs in chemistry: Biorenewable Deep Eutectic Solvent for Selective and Scalable Conversion of Furfural into Cyclopentenone Derivatives, Molecules, 2018; Green Synthesis of Privileged Benzimidazole Scaffolds Using Active Deep Eutectic Solvent, Molecules, 2019; Deep Eutectic Solvents as Effective Reaction Media for the Synthesis of 2-Hydroxyphenylbenzimidazole-based Scaffolds en Route to Donepezil-Like Compounds.  2020. L68-69
  • DESs as environmentally friendly and sustainable extraction solvents: Natural deep eutectic solvent as extraction media for the main phenolic compounds from olive oil processing wastes, Antioxidants, 2020. L70-72
  • Concerning the recycling and re-use of the DES post-reaction. This is again a remark of high interest. However, during the work-up of our highly viscous DES-based process, to extract products efficiently (up-coming manuscript) we had to “break” the DES´s H-bond network via aqueous solvation. Subsequently to allow an eased liquid-liquid extraction with the organic solvent for example. In this regard, re-using the components of our DES (e.g, Choline Chloride, Sorbitol…) entangled with unreacted substrates or unextracted products is seemingly doable at the lab scale but would represent a high challenge for industrial application. Challenge which would ultimately be defeated by the very low price of our DESs partners and the high price of purifications methods, lyophilization would be needed additionally to control accurately the water content necessary for the well-function of our process. To conclude, possibly a re-valorization of the used DES-containing mixtures seems foreseeable, specifically to our process. We tested before a method similar to the DES-recycling method applied in the paper from Di Gioia et al. 2018 but the mixing in our case was very unconvincing.

Nonetheless we are thankful for the great idea that would serve our next manuscript concerning the DSP optimization of the “2-in-1” process in which we could compare for example, the extraction with vs without solvation step.

Consequently, we added a paragraph on DES reusability and potential DES-waste treatment in our discussion. L312-318

Reviewer 2 Report

Review of Molecules Delavault manuscript

Overall, this is a nice manuscript.  It describes interesting research that is certainly better thought out and conducted that many similar enzymatic studies.  The care taken in characterization and quantification is certainly welcome.  I have three comments/questions.  The most important one is on page 7, where the data in Figure 4b do not make sense.  Why is the SL titer lower for 0.75 than the data for either 0.5 or 1.0?  This trend (or lack of trend) does not make sense.  Also, it is not clear if this data is from multiple replicates of reactions run under these conditions or from a single reaction under each set of conditions.  It would be nice if it was clearer the number of replicates (particularly since you talk about statistical significance).  And, if it was multiple replicates, what are the error bars?  That would be helpful.  The second one is on page 10 where you also mention recycling the immobilized catalyst.  Did you try this?  If not, I am not certain that this comment about line 272 is important (although it does not hurt).  Lastly, the English is not too bad, but it could use a little polish, below I have several of my own suggestions that will hopefully help.  Do note, though, that everything is clearly readable and understandable, so that is good.

I do like the work described in this manuscript and do recommend its publication with a few edits/clarifications.

Page 2 line 47 – replace nowadays with present

Page 2, line 48, replace problematics with problems (the same thing on line 56)

Page 2 line 49 – change revolves to revolve

Page 2, line 89 – change once to one

Page 3 line 106 – allowed the investigation of the impact

Page 3 line 113 – allowed the differentiation of the products

Page 4 line 136 – change few to little

Page 5 line 144 – remove the word “about”

Page 5 line 156 – To determine the optimal reaction

Page 9 line 226 – remove the words “to reach”

Page 9 line 227 – insert “to be reached” after at the tube scale

Page 9 line 229 – change sensible to sensitive

Page 9 line 232 – I would break this sentence up to be …the scalability of our process.  More investigation specific to this…

Page 10 line 252 – change esters to ester

Page 10 line 255 – change performances to performance

Page 10 line 268 – remove “albeit”

Page 10 line 269 – I would also break this sentence up - …value in titer of the product.  At least on the…

Page 10 line 273 – change remind to remember

Page 11 line 276 – change to rivaled the fuggered

Page 11 line 300 – change few to fewer

Page 11 line 308 – remove “notably”

Page 13 line 391 – change to “offer not only to the field of”

Page 14 line 420 – change to For this comparison, the reaction was

Page 14 line 449 – change to “filtered through a Buechner (sorry – you can leave the umlaut, I just cannot find it on my character map) funnel.”

Page 16 line 517 – Conclusions should be section 5

Page 16 line 527 – I would remove “actually their”

Author Response

We hereby thank the reviewer for his comments and suggestions.

The comments and suggestion are highly relevant for the overall betterment of the manuscript and for clarifications for potential future readers.

  • Concerning the data in Figure 4b, the lack of trend appeared at first misleading to us as well. Therefore, we considered a comparison between 0.25 M and 0.5 M being significant and a “safe” affirmation. Above 0.5 M however, we have clearly an inhibition linked to substrate concentration. In consequence we consider a comparison of concentrations strictly above 0.5 M being less significant despite what the statistical analysis tends to indicate. In further studies it would be therefore interesting (particularly for the optimization of the STR process) to “zoom” in this 0.5-1 M concentration range (with smaller incrementation) to define the trend less ambiguously. As a compromise and a sign of good faith, we highlighted this ambiguity in our results. L191-193
  • As stated in the subsection 4.10. of the Materials and Methods: “Results are presented as mean ± standard deviation (n = 3). Statistical analysis was performed by one-way ANOVA and Tukey test, results were considered significant if p-value was <0.05 if not stated otherwise.” Thus, error bars indicate standard deviation of the mean value obtained via three independent experiments running under the similar conditions stated. We added precision under the Figure 4 as well. L185-186
  • Concerning the recycling of the enzyme, we do have the full data set of the latter within the use of the described process. However, we would like to publish the re-usability of immobilized biocatalyst in a follow-up (currently on-going) study that would go beyond the simple of fact of stating “the enzyme can be re-used for X cycles and loses/retains X amount of activity”. Thus, we decided to mention it in the discussion because it must be present regarding justification of potential cost-efficiency of bio-catalyzed processes. To conclude we´d like to reserve this data to give an original view of re-used biocatalysts, in the framework of more innovative heating methods and alternative solvent systems.
  • All edits on the written English have polished further the language in the manuscript. Therefore, each of them have been considered and modified into the revised version of the manuscript.

Round 2

Reviewer 1 Report

The manuscript is well written and the corrections made by the authors ensure that the manuscript can be published in Molecules.
The only correction to be made is reference 14. The magazine is Molecules 2018 Jul 28; 23 (8): 1891. doi: 10.3390 / molecules23081891